# Elite Tennis Players Experiencing High-Arched Supination and Cuboids Dropped Foot Syndromes in Daily Normal Gait

**DOI:** 10.3390/ijerph19158897

**Published:** 2022-07-22

**Authors:** Tong-Hsien Chow, Chin-Chia Hsu

**Affiliations:** 1Department of Sports Science, R.O.C. Military Academy, Kaohsiung 830208, Taiwan; 2Department of Leisure Sport and Health Management, St. John’s University, New Taipei 25135, Taiwan; hill@mail.sju.edu.tw

**Keywords:** elite tennis players, plantar pressure distributions (PPDs), arch index (AI), high-arched supination, cuboids dropped

## Abstract

Many studies have focused on the plantar pressure characteristics of specific movements and footwork in tennis. However, little research has been conducted for exploring the foot characteristics among tennis professionals’ daily habitual paces. This study aims to examine the pressure profiles associated with foot posture and balance abilities of elite tennis players during normal gait to understand how foot loading patterns result from habitual paces that may be derived from intensive tennis training and competition. A cross-sectional comparative study is conducted on 95 male college elite tennis players (mean age: 20.2 ± 1.2 years) and 100 male recreational tennis players (mean age: 19.8 ± 0.9 years). Bipedal plantar pressure distributions (PPDs) associated with arch index (AI) and centers of gravity balance are explored through the plantar pressure device. The foot posture is estimated to determine the rearfoot postural alignment. During the midstance phase of walking with a normal gait, the bipedal AI values of the elite group are significantly lower, indicating that they have high-arched feet. Additionally, the elite group experienced higher PPDs at the lateral regions of their longitudinal arches and heels and relatively lower PPDs at the medial portions of both feet. Rearfoot postural alignment resonance analysis of the PPDs suggests that the elite group experienced foot supination associated with cuboid dropped. Moreover, the right foot bears heavier centers of gravity balance in the present study. The elite tennis players in the study are categorized as having high-arched supination with cuboids dropped when performing daily habitual paces. This finding warrants further investigation into the correlation between possible injuries and daily habitual paces that may result from tennis’ intensive training and competition.

## 1. Introduction

Tennis is a high-intensity anaerobic exercise [1] characterized by repeated and explosive motions so that it involves multiple muscle groups during different strokes [2]. The most common cause of injuries reported among tennis players was overuse [3]. Since tennis involves skating and running, the lower extremities are primarily affected by repetitive sprinting overloads [4]. Lower extremity injuries were more common (48%) than upper extremity injuries (29%) among all reported injuries [5]. Studies have indicated that lower limb injuries are most common in the tennis discipline, particularly footwear, ankle sprains, meniscal knee injuries, knee tendinopathy, hip injuries, and tennis leg [6]. Plantar fasciitis, tennis toe, and stress fractures are common foot injuries among tennis players [7]. A sudden and unexpected ankle dorsiflexion or severe dorsiflexion of the plantarflexed foot are both common causes of acute injuries in tennis [8]. Therefore, foot problems are an unavoidable injury region in the tennis discipline [9].

Foot structure has a major impact on dynamic foot function [10]. Yet, plantar loads may have a link between foot structure, lower limb function, and balance ability [11]. Plantar pressure distribution may serve as an accurate measurement of plantar loads, abnormal gaits, lower limb alignment, podiatric severity, and rehabilitation condition [12] as well as confirm the accuracy of movement [13,14]. Many studies have also demonstrated that a change in foot loading pattern is accompanied by alterations in the structure of the longitudinal foot arch [15,16,17]. The medial longitudinal arch (MLA) of the foot provides adequate elastic forces and twisting forces to absorb the ground reaction force during activities, thus attenuating impact and delaying fatigue [18]. The height of the MLA can affect the alignment of the lower limbs. An elevated MLA affects the foot’s supination alignment, whereas a lowered one is associated with its pronation [19]. Therefore, abnormal MLA function can affect lower extremity biomechanics and contribute to podiatric diseases [20]. For example, an athlete with an abnormally supinated foot may be more prone to metatarsal stress fractures because a more rigid foot is less able to attenuate weight-bearing stresses [20]. Furthermore, a supinated foot can also be reminiscent of the cuboid syndrome, which is most likely to occur in athletes who engage in sports such as ballet, basketball [21], rugby [22], tennis, and running [23]. Since the foot and ankle of a human body provide weight bearing and weight transfer capabilities during activities, identifying plantar loading information, such as the relationship between location and magnitude, could provide insight into the fine structure and function of the foot in the gait cycle [14,24]. The measurement of plantar pressure and gait analysis is widely accepted as valuable biomechanical parameters for quantifying human gait, which can provide useful information about foot function and aid in the development of more effective prevention and intervention strategies [25,26].

Previous studies involving tennis discipline generally focused on the effect of the tennis serve, footwork or related specific movements, and the court surface on players’ plantar pressure distribution [27,28,29], e.g., male tennis players experienced higher peak and mean pressure as well as a maximum force for the heel region during hard-court competition [30]. Kornfeind et al. went further and proposed that tennis players performing accelerating and decelerating movements on clay exhibited higher rearfoot plantar pressure distribution [31]. Girard et al. found that peak pressure and mean pressure were higher in the lateral forefoot during the tennis serve than in the stationary serve, but lower in the medial forefoot and middle foot [27]. However, these studies do not accurately reflect the plantar pressure distribution experienced by tennis players in their daily lives, and limited evidence has been reported regarding plantar pressure during normal gait. Given the above context, the study is based on a series of past experiences exploring the correlation between static and dynamic foot pressure profiles and lower extremity pain profiles in specific elite athletes [21,22,32,33]. On the basis of these findings, this study aimed at investigating the plantar pressure characteristics and foot posture of tennis players during their natural gait. It is worth further exploring because variations in foot arch index (AI), plantar pressure distributions (PPDs), the balance of centers of gravity, and rearfoot postural alignment may be associated with mature experiences of tennis skills in college elite tennis players. We hypothesized that the elite tennis players in this study were classified into high arch types and that their PPDs were particularly concentrated in the lateral regions of the midfoot and rearfoot. Moreover, the plantar pressure characteristics and rearfoot posture may correlate with the features of supinated and cuboid dropped foot patterns.

## 2. Materials and Methods

### 2.1. Participants

A cross-sectional comparative study of Taiwanese college and university students was conducted during the non-competition period. Participants recruited in the present study were categorized into the following two groups: 95 elite tennis players (the elite group) and 100 healthy age-matched recreational tennis players (the recreational group). Participants in the elite group were considered first-division tennis athletes with a preference for right-handedness and had at least four consecutive years of tennis training and experience competing in the National College Tennis Elite Tournament, National College Tennis Championship, and National Intercollegiate Athletic Games as well as registered in the Chinese Taipei Tennis Association (CTTA). They were selected from among 132 male elite tennis athletes who were recruited from National Taiwan Sport University, University of Taipei, National Taiwan University of Sport, National Taipei University of Business, Chinese Culture University, Tamkang University, Feng Chia University, Da Yeh University and Tainan University of Technology in Taiwan. According to statistics, the participants in the elite group were all between the ages of 19 and 21 and there was a 28% drop-out rate during recruitment of the eligible elite tennis athletes due to the following: (1) the absence rate; (2) the physician’s certificate that they previously fractured or had surgery; (3) the fact of taking or having taken anti-inflammatory analgesics in the past month. Their physical ability training schedules include stretching exercises and aerobic running from 2 PM to 4 PM. The tennis tactical training sessions were scheduled from 5 PM to 7 PM on a daily basis for four to five days a week.

For the recreational group, there were 100 eligible participants who were selected from 147 recreational tennis players. They have recreational tennis experiences for at least 2 years and played at least 2 days per week at a tennis court within six months before this study. An approximate 32% drop-out rate was observed during the recruitment of the eligible recreational group due to the following reasons: (1) the absence rate; (2) having trained professionally in other sports disciplines; (3) having a physician’s certificate of past fractures or surgery.

Elite and recreational groups were both recruited from a relatively homogenous population. Participants in two groups had varying training intensities, training patterns, workout schedules and experienced different levels of competition. Participants in both groups were excluded if they had had any lower limb trauma, fracture, and surgery within the past six months, or if they had any musculoskeletal disorders such as calcaneal spurs, rheumatoid arthritis, or neuropathies. All participants’ age, height, weight, and body mass index (BMI) were recorded during the study. Table 1 summarizes the anthropometric characteristics of the study groups. Significant differences between the two groups in the dimensions of height, mass, and BMI after the two-group Student *t*-tested with a 95% confidence level.

### 2.2. Procedure

The foot arch index (AI), plantar pressure distributions (PPDs), and balance of centers of gravity of all participants were conducted after anthropometric characterization. The parameters of these experiments were recorded simultaneously during walking conditions. The results were measured and analyzed using the JC Mat optical plantar pressure analysis device. Then, the rearfoot postural alignment was assessed in the final study procedure. All experiments in this study were conducted in accordance with guidelines of the research ethics committee at National Taiwan University and with the recommendations of the Declaration of Helsinki.

### 2.3. Instruments

Study participants were assessed using the JC Mat optical plantar pressure analysis device (View Grand International Co., Ltd., New Taipei City, Taiwan) and built-in FPDS-Pro software to determine the arch index (AI), plantar pressure distributions (PPDs), and balance of the foot centers of gravity [34]. The device has been proven repeatable and reproducible in previous studies [21,22,32,33]. The JC Mat uses the same principle and technology as the Harris footprint measurement. The main characteristics of the device are as follows: (1) foot characteristics are easily and effectively recognized; (2) PPDs and footprint images are consistent with results of weight calibration experiments; (3) for measuring plantar pressure, there are 25 sensors per square centimeter on each side of the JC Mat (32 × 17 cm); (4) the sensitive pressure sensing system with a wide working area is able to display and mark the delicate plantar pressure image with fine dots; (5) the barefoot images, as well as the pressure distribution of footprints, can be captured immediately; (6) with built-in FPDS-Pro software it is possible to analyze certain parameters such as the AI and PPD values, balance of the centers of gravity, and toe angle.

### 2.4. PPDs Assessment

For consistency and reliability, the experiments were conducted three days before and after their regular training, and one month before and after competitions. Each experiment was conducted every Monday and Thursday from 2 to 5 PM. All participants were required to undergo anthropometric measurements to record their basic physiological characteristics. In order to capture the preliminary results of the walking footprint, the participants were required to complete short walks on the trail, following the following procedure:Participants were asked to first roll up their trouser legs above the knees to ensure that clothing did not restrict their movements;Barefoot stand in a natural posture, with arms hanging vertically at sides;Each participant was instructed to walk barefoot at their self-comfort speed, following their accustomed gait and pace [35,36,37] along the 4-m-long JC Mat built-in walkway to its end;Turn around at the end, and immediately walk to the starting point with a natural gait;Each participant will need to practice the above steps beforehand, and then enter the experiment after confirmation;Upon entering the experiment, each participant was instructed to perform several walking trials until at least three take-off paces were correctly acquired with the left and right foot (i.e., a single foot strikes the sensing cushion marked with the specific measurement range of the JC Mat).

After receiving the multiple walking data, the experiment was terminated by pressing the stop button. The preliminary data was automatically saved when the close button was pressed.

### 2.5. PPDs Data Analysis

The AI and PPD values were measured and analyzed from five footprint regions and sub-regions of both feet. The digital images of footprints from the walking trail were managed using a computer program (FPDS-Pro software) integrated into the JC Mat. During the analysis, the software generates the first line (a vertical line) on the footprint image. It then generates tangent lines to the foremost and rearmost end of the footprint, excluding the toes, from the tip of the second toe to the center of the heel. Meanwhile, the software automatically forms four parallel lines perpendicular to the vertical line. Therefore, the footprint can be equally divided into six distinct subregions (subregions 1, 2, 3, 4, 5, and 6), which were identified through segmentation. They were defined as (1) the lateral metatarsal bone (LM), (2) the lateral longitudinal arch (LLA), (3) the lateral heel (LH), (4) the medial metatarsal bone (MM), (5) the medial longitudinal arch (MLA), and (6) the medial heel (MH), respectively. In addition, these six subregions could be further combined with each other into the five regions. Three regions of A, B, and C in the front and rear directions, and two regions of D and E in the left and right directions. On the illustrated footprint, regions A, B, and C correspond to the forefoot (subregions 1 and 4), midfoot (subregions 2 and 5), and rearfoot (subregions 3 and 6), respectively. Regions D and E correspond to the lateral (subregions 1, 2, and 3) and medial foot (subregions 4, 5, and 6) (Figure 1). The study referred to Cavanagh and Rodgers proposed a formula for calculating AI as a ratio of the region of the middle third of the footprint divided by the region of the entire footprint except for the toes, i.e., AI = B/(A + B + C). As defined by Cavanagh and Rodgers, an AI below 0.21 indicates a high arch, an AI between 0.21 and 0.26 indicates a normal arch, while higher than 0.26 indicates a flat arch [38].

### 2.6. Rearfoot Postural Alignment Assessment

During the experiment on the rearfoot postural alignment, each participant was instructed to static stand on a 30 cm height platform and keep their feet spaced naturally (around 12–15 cm). After standing still and stable on the platform, the instructor used a digital camera to capture each participant’s posterior view of rearfoot postural alignment image. According to the method used to calculate the rearfoot static angle [39], the rearfoot of both feet stand on the same horizontal line, and the anatomical points are as follows: (1) the posterior calcaneal tuberosity, (2) the second point above the center of the calcaneus, and (3) the lower third of the leg. The Biomech 2019-postural analysis software was applied to connect three points and automatically generate two intersecting lines (Loran Engineering SrL, Bologna, Italy). Additionally, the software automatically draws the first standard straight line (a solid line) of the lower extremity that originates from the lower third of the leg to the center of the calcaneus. The second flip angle line (a dotted line) of the lower extremity was extended from the posterior tubercle of the calcaneus to the center of the calcaneus. As a result of the intersection between those two straight lines, angles were classified as normal foot (0° to 5°), varus (<0°), and valgus (>5°) [40].

### 2.7. Statistical Analysis

Study participants’ age, height, weight, and BMI values were measured and described using descriptive statistics, and expressed as mean ± standard deviation (SD). The AI values, balance of the centers of gravity, and three regional and six subregional PPDs of study groups were compared using the independent sample *t*-test. A significant statistical difference was defined as *p* < 0.05. and *p* < 0.01. Statistical analyses in the study were conducted using the software program (IBM SPSS advanced statistics version 20.0, SPSS Inc., Chicago, IL, USA).

## 3. Results

### 3.1. Bipedal Arch Index

The bipedal AI analysis revealed that the average value of the recreational group was within the normal range (from 0.20 to 0.26), whereas the elite group’s AI value was significantly lower than the recreational group, meaning the arch height was relatively higher in the elite group (Table 2).

### 3.2. The PPDs of the Five Regions of Both Feet during the Midstance Phase of Walking

In the present study, the PPDs were expressed as percentages of the relative loads. Plantar pressure distributions beneath the bipedal forefoot, midfoot, rearfoot, lateral, and medial foot during the midstance phase of walking were shown in Table 3. The elite group’s PPDs were found to be significantly higher in the midfoot regions of both feet (Left foot: 12.23% ± 11.58%; Right foot: 12.59% ± 11.76%; *p* < 0.01), the rearfoot (17.82% ± 9.40%; *p* < 0.01) and lateral foot (24.03% ± 3.53%; *p* < 0.05) regions of the right foot. Yet, the PPDs were relatively lower in the forefoot regions of both feet (Left foot: 20.86% ± 3.87%; Right foot: 19.59% ± 3.30%; *p* < 0.05), the rearfoot region (16.91% ± 8.97%; *p* < 0.01) of the left foot and the medial foot region (9.30% ± 7.38%; *p* < 0.01) of the right foot.

### 3.3. The PPDs of the Six Subregions of Both Feet during the Midstance Phase of Walking

The results showed that during the midstance phase of walking with normal gait, the PPDs in the elite group showed significantly greater at the lateral longitudinal arches (Left foot: 23.59% ± 2.92%; Right foot: 24.19% ± 2.44%; *p* < 0.01) and the lateral heels (Left foot: 25.37% ± 3.66%; Right foot: 26.84% ± 2.73%; *p* < 0.01) of both feet, whereas were found to be relatively lower at the medial metatarsals (Left foot: 20.67% ± 3.93%; Right foot: 18.11% ± 3.18%; *p* < 0.05), the medial heels (Left foot: 8.45% ± 1.96%; Right foot: 8.81% ± 2.42%; *p* < 0.01) and the medial longitudinal arches (Left foot: 0.87% ± 0.35%; Right foot: 1.00 ± 0.33%; *p* < 0.05) of both feet (Table 4) as compared with the recreational group.

### 3.4. The Balance of the Centers of Gravity

The balance assessment was determined as a percentage of the centers of gravity. The results showed that during the midstance phase of walking with a normal gait, the elite group experienced higher centers of gravity on the right foot (51.52% ± 4.09%; *p* < 0.01) and lower on the left foot (48.48 ± 4.09%; *p* < 0.01) as compared to the recreational group. Moreover, no significant differences were observed between both feet within the respective groups (Table 5).

### 3.5. Bipedal Rearfoot Postural Alignment

Static angles of the bipedal rearfoot postural alignment were measured in angles and expressed in degrees (deg.). The results showed that the elite group had significantly lower values when compared with the recreational group (Table 6).

### 3.6. Characteristics of Footprints

Footprint images of each homogenized representative subject were determined by averaging results from plantar pressure analysis within the respective group. The footprint characteristics of the elite group displayed a certain tendency for supination and were accompanied by higher plantar pressure distributed to the bipedal lateral longitudinal arches and the lateral heels. The worn-out areas of a sneaker sole frequently worn by the representative elite subject correspond to the regions of higher-pressure distributions on the bipedal plantar surface (Figure 2).

## 4. Discussion

Measurement of plantar pressure has been maturely applied in examining foot abnormalities [17], diagnosing specific podiatric problems [41], assessing lower extremity function [11], studying gait behavior [42], and developing for orthotic insole or foot orthoses design [43]. Evaluation of individual plantar pressure distribution upon static standing [21,32,33] or while performing specific footwork may provide essential insights for studying the effects of movements on plantar loading [27,28,29]. Particularly for athletes, plantar pressure measurement is considered to be a convenient and efficient tool for analyzing foot, ankle, and lower extremity motor behaviors [21,22,32,33]. In recent years, the measurement has been commonly used to investigate the effect of specific movements, such as serving, striking, footwork, and pedaling on the court surface, on the distribution of plantar pressure. Despite this, however, few studies have been conducted for examining the experienced natural gait of a specific disciplined athlete to determine the plantar pressure characteristics and foot posture during their daily normal walking. In order to understand the differences between the present study and the previous literature, the study recruited tennis-specialized athletes to analyze their plantar pressure distribution, centers of gravity, and foot posture during the midstance phase of walking. The features of plantar pressure profiles, centers of gravity balance, and foot posture of the natural gait of daily living in elite tennis players appear to be derived from their mature experiences with tennis skills. 

First of all, the bipedal AI values within the respective groups were considerably symmetrical to each other. The recreational group had a normal arch type. Yet, the average AI value of the elite group was significantly lower. The result was similar to that of previous studies of college-level male elite basketball players [21] and elite sprinters [32] and they were classified as having high-arched feet. The research on the performance of athletes with high arches found that short-distance runners with high-arched feet had better dynamic balance and speed compared to runners with low and neutral arch feet [44]. High-arched feet are favorable conditions for sprinters in competition [45]. These studies suggest that athletes with high arches could be potential candidates for activities such as sprinting and dynamic balance activities [44,45]. According to research, athletes with high-arched feet have the potential to sprint and balance dynamically. As mentioned above, the elite tennis players in the present study were more specialized in different technical skills in competitions, such as accelerations and decelerations, short sprints with sudden stops in various directions. As a result, these dominant conditions for movement behavior may be attributed to the contribution of the high-arch type.

As a consequence of bipedal plantar pressure distributions during natural gait, the elite group’s relative loads were mainly distributed to the midfoot regions of both feet and the right rearfoot and lateral regions, while the relative loads on the forefoot regions of the feet were symmetrically lower. The six subregional PPDs results further showed that the elite group experienced greater relative loads on the bipedal lateral regions of the longitudinal arches and heels. The medial portions of both feet in the elite group experienced symmetrical lower relative loads. The results were consistent with the rearfoot postural alignment in the present study, indicating the elite tennis group had an elevated rearfoot varus as compared to the recreational group. Such results can be referred to past studies by Martinez Nova et al. that assessed the foot posture index (FPI) of athletes and concluded that handball players had a greater tendency toward foot supination [46]. De Cesar Netto et al. went further and pointed out that NBA professional basketball players appear to have a normally aligned hindfoot with a tendency toward a varus morphotype and a high-arched foot [47]. These cases resonate with an argument proposed by García-Pinillos et al. concerning the effects of an elevated longitudinal arch on supination alignment [19]. There has been earlier research demonstrating that people with high arches or supinated feet had larger center-of-pressure (COP) excursions than people with normal or pronated feet [48]. Healthy young adults with supinated feet had a significantly higher average speed of COP [49]. An individual with a supinated foot may provide a rigid lever for propulsion of walking [50]. The increase in supination may be related to the strength and activity of the tibialis anterior as well as relatively weak peroneal muscles, causing the movement to be supinated [51]. In addition, the relationship between supinated feet and postural stability was mentioned in previous literature. An earlier study by Cote et al. indicated that participants with supinated feet had better postural stability during single leg stance with their eyes open than those with pronated feet, whereas there was no difference in postural stability between groups with supinated and neutral feet [52]. Angin et al. went further and argued that participants with supinated feet had a significantly lower center of gravity sway velocity during unilateral stance with their eyes closed compared to participants with normal or pronated feet. The study suggested that people with supinated feet were better able to maintain their postural stability [53]. As mentioned above, a recent study conducted by Beelen et al. found that a positive effect existed in the relationship between supinated feet and postural stability [54]. 

The results of the present study showed that the elite group had a higher center of gravity on the right foot and lower on the left foot than the recreational group. Nonetheless, there were no significant differences between the bipedal centers of gravity within the respective groups. The findings of this study to some degree echoed that previous research showed that participants in the elite group have a symmetrical balance on centers of gravity on both feet. As for the situations, the elite group had a heavier center of gravity on the right foot and lower on the left foot than the recreational group, which may be explained as a feature of the exercises of regular training and competition in tennis athletes who prefer right-handedness. As for the relationship between hand and foot preferences, the study by Barut et al. indicated that among people with right-handedness, approximately 75.5% preferred the right foot [55]. Packheiser et al. also conducted meta-analyses to confirm the strong connection between right-handedness and right-footedness, while the link between left-handedness and left-footedness is less pronounced [56]. The relationship between handedness and footedness has been examined in several studies and was found to be correlated positively, indicating a common biological mechanism causing lateralized motor function [57,58]. 

As for the result of footprint images, the study screened out a homogenized representative subject through the average of the plantar pressure analysis data within the respective group. We illuminated that the elite tennis players had high-arched supination and dropped cuboid footprints. The cuboid syndrome usually results from either low arches or high arches [21]. Abnormal foot arches may cause disproportionate pressure on the cuboid bone and contribute to fatigue in the musculoskeletal system of the lower limbs [21]. Such results of the study specifically echoed the theory that a supinated foot could be related to symptoms of cuboid subluxation, and it is most likely to occur in athletes specializing in sports disciplines such as ballet, basketball [22], rugby [22], tennis, and running [23]. The common characteristic of these sports is that athletes change directions rapidly in quick movements, with the feet experiencing large forces of tension and torsion. Consequently, the muscles attached to the cuboid bone are overused and tightened, resulting in increased pressure on the joints and attachments of the cuboid bone. In some instances, the repetitive pressure of regular exercise training is unavoidably applied to the cuboid bone, which could damage the surrounding ligaments or joint capsule, causing the support around the cuboid bone to slacken, thereby causing the cuboid bone to dislocate [21]. Considering the above descriptions, a previous study specifically described this phenomenon that an individual with a high-arched foot or an anterior equines deformity with a plantarflexed lateral column may cause them to walk in a more supinated position, which could increase lateral pressure on the midfoot. This may make cuboid syndrome more prevalent. Therefore, a certain degree of increased loading on the lateral portion of the foot may present a traceable feature for cuboid syndrome and supinated foot.

The present study was limited by the fact that the issues of preferred/non-preferred legs of the participants were not considered due to the fact that only participants with a preference for right-handedness were included in the study. Therefore, further studies should be conducted to understand whether elite tennis players have a dominant leg based on a larger number of research samples and to investigate how dominant legs may influence plantar pressure distribution in particular. Additionally, since the results of the walking plantar loads in the dynamic tests were obtained while the participants walked at their own pace on the device, differences in the participants’ walking speeds are bound to exist during the process of testing. As a result, the data collected by the researchers may be affected to some extent. Nevertheless, there is still limited evidence examining the characteristics of plantar pressure distributions and centers of gravity balance as well as foot posture during an elite tennis player’s normal gait on a daily basis. 

The study considering the differences in the foot loading patterns of regular walking versus previous cases of specific tennis footwork is critical to understanding the injury mechanisms resulting from daily habitual paces that may be derived from tennis intensive training and competition. Findings from the study could offer insight into the dynamic plantar pressure distribution associated with centers of gravity balance and foot posture for Taiwanese college and university elite tennis players. The results of the plantar pressure profiles and foot posture of elite tennis players may prove useful for the development of prevention strategies for rearfoot varus and cuboid syndrome in tennis players, such as core muscle group stretching, insole correction, appropriate footwear, and sneaker selection.

## 5. Conclusions

Findings of the present study showed that during the normal gait of daily life, elite tennis players presented high-arched supination along with a higher lateral column of plantar loads. The situation was accompanied by dropped cuboids and experienced higher centers of gravity balance on the right foot. The results could provide insights into the features of plantar pressure profiles and foot posture for elite tennis players.

## Figures and Tables

**Figure 1 ijerph-19-08897-f001:**
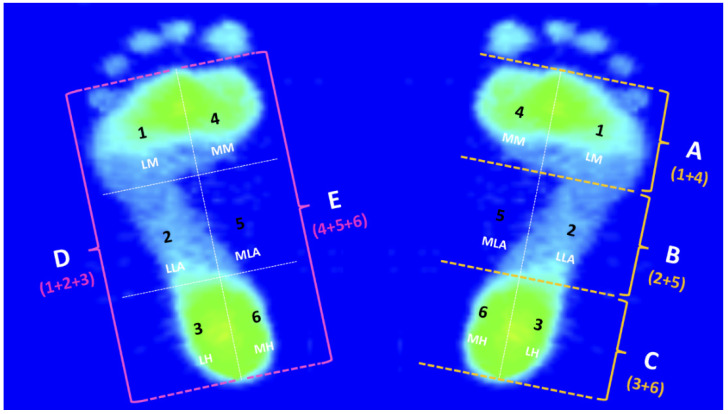
Footprint diagram for the five regions and the six subregions. The six subregions are numbered sequentially from 1 to 6 and their abbreviations are as follows: (1) LM, lateral metatarsal bone; (2) LLA, lateral longitudinal arch; (3) LH, lateral heel; (4) MM, medial metatarsal bone; (5) MLA, medial longitudinal arch; (6) MH, medial heel. The five regions A, B, C, D, and E correspond to the forefoot (subregions 1 and 4), midfoot (subregions 2 and 5), rearfoot (subregions 3 and 6), lateral (subregions 1, 2, and 3) and medial (subregions 4, 5, and 6) foot, respectively.

**Figure 2 ijerph-19-08897-f002:**
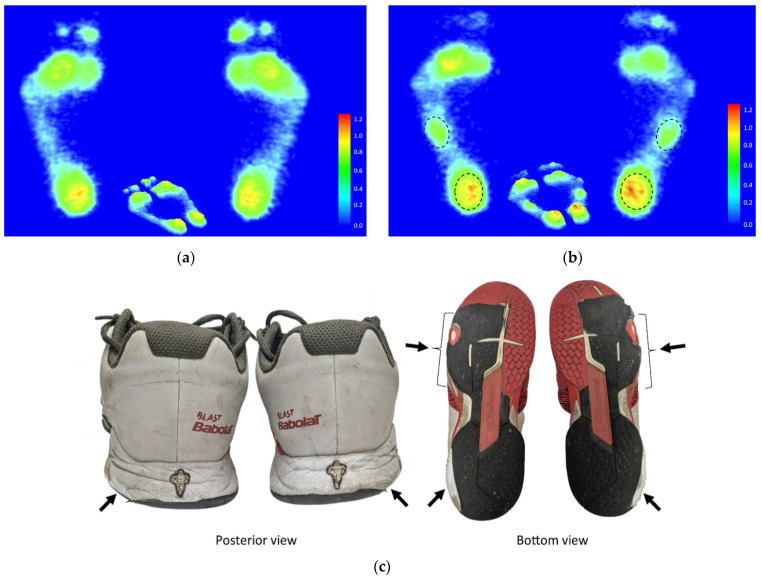
The footprint of each representative subject of the recreational group (**a**) and the elite group (**b**) was determined by plantar pressure homogenized results analysis. The worn-out areas of a sneaker sole (**c**) frequently worn by the representative subjects of the elite group. Black dash circle indicates the areas of higher pressure. Black arrow indicates the areas of worn-out sneaker sole.

**Table 1 ijerph-19-08897-t001:** Description of the anthropometric characteristics of the groups.

Characteristic	Recreational Group ^1^ (n = 100)	Elite Group ^2^ (n = 95)
Age (years)	19.8 ± 0.9	20.2 ± 1.2
Height (cm)	170.7 ± 4.2	176.0 ± 5.3 *
Mass (kg)	67.7 ± 3.7	71.0 ± 5.7 *
BMI (m/kg)	23.2 ± 0.6	22.9 ± 0.9 *
Tennis training experience (years)	2.7 ± 0.6	4.9 ± 0.9

Abbreviation: BMI, body mass index (calculated as the weight in kilograms divided by the square of the height in meters). Note: Values are given as mean ± SD. * *p* < 0.05. (student-*t* test, 2-tails). ^1^ Healthy eligible recreational tennis players (the recreational group) were college and university students who preference for right-handedness. ^2^ Elite tennis players (the elite group) were age-matched and considered to be first-division tennis athletes with a preference for right-handedness and had at least four consecutive years of tennis training and experience competing in the National College Tennis Elite Tournament, National College Tennis Championship, and National Intercollegiate Athletic Games as well as registered in the Chinese Taipei Tennis Association (CTTA).

**Table 2 ijerph-19-08897-t002:** Bipedal arch index of the foot.

	Recreational Group (n = 100)	Elite Group (n = 95)	*p* Value ^1^
Left foot	0.21 ± 0.06	0.18 ± 0.08	0.011
Right foot	0.21 ± 0.04	0.18 ± 0.06	< 0.01

Note: Data are given as mean ± SD. ^1^ *p* values were determined by the independent sample *t*-test between the recreational group and the elite group.

**Table 3 ijerph-19-08897-t003:** Plantar pressure distributions of the five regions of both feet during the midstance phase of walking.

Five Regions	Recreational Group (n = 100)	Elite Group (n = 95)	*p* Value ^1^
Left foot			
Forefoot (%)	21.67 ± 3.11	20.86 ± 3.87	0.018
Midfoot (%)	11.37 ± 10.48	12.23 ± 11.58	<0.01
Rearfoot (%)	16.97 ± 4.72	16.91 ± 8.97	<0.01
Lateral foot (%)	21.43 ± 3.60	23.34 ± 3.90	0.328
Medial foot (%)	11.91 ± 8.46	9.99 ± 8.55	0.671
Right foot			
Forefoot (%)	21.08 ± 2.66	19.59 ± 3.30	<0.01
Midfoot (%)	11.91 ± 10.66	12.59 ± 11.76	<0.01
Rearfoot (%)	17.03 ± 5.16	17.82 ± 9.40	<0.01
Lateral foot (%)	21.42 ± 3.16	24.03 ± 3.53	0.018
Medial foot (%)	11.94 ± 8.44	9.30 ± 7.38	<0.01

Note: Data are given as mean ± SD. ^1^ *p* values were determined by the independent sample *t*-test between the recreational group and the elite group.

**Table 4 ijerph-19-08897-t004:** Plantar pressure distributions of the six subregions of both feet during the midstance phase of walking.

Six Subregions	Recreational Group (n = 100)		Elite Group (n = 95)	
	Left Foot (%)	Right Foot (%)	Left Foot (%)	Right Foot (%)
Lateral Metatarsal bone (LM)	22.37 ± 3.04	21.16 ± 2.70	21.06 ± 3.82	21.07 ± 2.72
Lateral Longitudinal Arch (LLA)	21.37 ± 4.31	22.37 ± 2.54	23.59 ± 2.92 ^2^	24.19 ± 2.44 ^2^
Lateral Heel (LH)	20.71 ± 2.96	20.72 ± 3.87	25.37 ± 3.66 ^2^	26.84 ± 2.73 ^2^
Medial Metatarsal bone (MM)	20.97 ± 3.03	21.01 ± 2.62	20.67 ± 3.93 ^2^	18.11 ± 3.18 ^1^
Medial Longitudinal Arch (MLA)	1.37 ± 0.42	1.46 ± 1.17	0.87 ± 0.35 ^2^	1.00 ± 0.33 ^1^
Medial Heel (MH)	13.36 ± 3.04	13.29 ± 3.38	8.45 ± 1.96 ^2^	8.81 ± 2.42 ^2^

Plantar pressure distributions of the six subregions of both feet are represented as mean ± SD and *p*-values were determined by the independent sample *t*-test ^1^
*p* < 0.05, ^2^
*p* < 0.01, significant differences between the recreational and elite group.

**Table 5 ijerph-19-08897-t005:** The Balance of the Centers of Gravity for the Elite Tennis Players.

	Recreational Group	Elite Group	*p* Value ^1^	*p* Value ^2^	*p* Value ^3^
Left foot (%)	49.03 ± 3.09	48.48 ± 4.09	<0.01	1.000	1.000
Right foot (%)	50.97 ± 3.09	51.52 ± 4.09	<0.01	--	--

The percentage of centers of gravity of both feet are represented as mean ± SD and ^1^ *p* values were determined by the independent sample *t*-test between the recreational (n = 100) and elite group (n = 95). ^2^ *p* value was determined between both feet of the recreational group. ^3^ *p* value was determined between both feet of the elite group.

**Table 6 ijerph-19-08897-t006:** The bipedal rearfoot postural alignment for the elite tennis players.

	Recreational Group (n = 100)	Elite Group (n = 95)	*p* Value ^1^
Left foot (deg.)	2.42 ± 2.84	−2.13 ± 3.57	0.013
Right foot (deg.)	1.74 ± 3.01	−2.74 ± 3.14	0.004

Static angles of rearfoot postural alignment of both feet are represented as mean ± SD and ^1^ *p* values were determined by the independent sample *t*-test between the recreational and elite group.

## Data Availability

The datasets generated and/or analyzed during the current study are available from the corresponding author upon reasonable request.

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
