# Peer review of "Elite Tennis Players Experiencing High-Arched Supination and Cuboids Dropped Foot Syndromes in Daily Normal Gait"

_ijerph, 2022, doi:10.3390/ijerph19158897_

Round 1

Reviewer 1 Report

I would like to congratulate the authors to this excellent study.

Specific comments:

i) Abstract: include values of the main results, and comparisons;

ii) Introduction:

1) what does the term "demanding sports" means? (line 31);

2) what kind of "loading" (line 35);

3) Which are the assumption supporting the hypothesis about foot alterations and possible correlations (lines 72-76). Please, give further supports to expecteded findings.

iii) Methods

4) Revise the terms for arm position (line153);

(iv) Results:

5) absolute PPD values need to be inserted (from line 242);

6) Table 4, 5 and 6: what are the units of the measurements?

(v) Conclusion:

7) do not revise the literature again (lines 298 - 308)

8) do not revise the objectives (lines 308 - 311)

9) highlight and support the main findings in the first paragraph.

10) avoid large paragraphs, as that from line 329 to 397. Please, split it.

Author Response

Dear Reviewer,                      

We would like to resubmit our revised manuscript, entitled “Elite Tennis Players Experiencing High-Arched Supination and Cuboids Dropped Foot Syndromes in Daily Normal Gait” to the Special Issue (SI: Tennis and Padel: Performance and Health) of International Journal of Environmental Research and Public Health as an Article paper for possible evaluation.

We appreciate the reviewers’ constructive suggestions and comments on my resubmitted manuscript (ID: ijerph-1773592). The suggestions and comments are helpful for improving our manuscript. The title of our revised manuscript has been modified based on the Reviewer’s comments.

We are resubmitting the revised version of the manuscript with my responses to the suggestions and comments by the reviewers. Our responses to each suggestion and comment are presented in blue texts with a grey background color in the revised manuscript.

Correspondence about the paper please be directed to Tong-Hsien Chow, at the following address, fax number and e-mail address. I confirm that this manuscript is original, has not been published before and is not currently being considered for publication elsewhere.

Thank you very much for your consideration of this manuscript.

Sincerely yours,

Tong-Hsien Chow, Ph.D.

Associate Professor

Director of Department of Sports Science, R.O.C. Military Academy

Address: No.1, Wei-Wu Rd., Fengshan Dist, Kaohsiung 830208, TAIWAN (R.O.C.)

Tel: 886 7-7462151

Reviewer 2 Report

Dear author, first of all thank you for your submission.

The article you present is well structured, however I believe that there are some specific points that can be improved.

Title: I believe a title is a little confusing in the form it is in right now. Perhaps you can consider the change.

Line: 63-76 Dear author, it would be important to mention precisely the purpose of the study. It highlights the gaps in the hypotheses but does not clearly report the purpose of the study.

Author Response

(The authors gave the same response as above.)

Reviewer 3 Report

This study aimed to examine the pressure profiles associated with foot posture and balance abilities of elite tennis players during normal gait to understand how foot loading patterns result from habitual paces that may be derived from tennis intensive training and competition. The paper is The topic is quite interesting. Research is conducted in Taiwan and presumably the authors are not native English speakers, the quality of the English language is not sufficient. My first impression is that the paper needs a thorough proofreading and copyediting. I recommend the authors to have their manuscript reviewed by a native English language speaker.

Abstract: The abstract does not follow the editorial standard indicated for IJERPH: “The abstract should follow the style of structured abstracts, but without headings”; therefore, remove the words: Introduction, Methods, Results and Conclusions.

Abstract: "Considering that previous literature focusing almost exclusively on plantar pressure characteristics of specific movements and footwork in the tennis discipline." What means this sentence? Please revise it. The english grammar is wrong.

Abstract: Distribution of gender and age should be included in Abstract.

In Introduction many parts of the literature review only simply put previous studies together without themes. So, readers couldn't get a clue of focus. What do authors want to tell readers? It is very confusing. 

 You should follow the style of reference in the text. The style is wrong.

Introduction: pag 2 line 63-65  "According to previous studies involving tennis discipline focused on the effect of the tennis serve, footwork or related specific movements, and the court surface on players’ plantar pressure distribution." Again, what means this sentence??? According bla bla....and what??

Introduction: pag 2 line 72-75 please reformulate the aim of this study and then present your hypotheses. 

What the version of SPSS is used in this research?

A zero should not be inserted before a decimal fraction when the number cannot be greater than 1. For example, p < 0.05 should be written as “p < .05.” Continues in the same way!

You have to add the section Procedure after Participants.

Please include new separate paragraph discussing limitations and future research of the study.

Conclusions should be based on the results, clearly stated and rewritten.

Author Response

(The authors gave the same response as above.)

Round 2

Reviewer 3 Report

The revised manuscript is suitable for the publication